# Subepithelial Stromal Cells: Their Roles and Interactions with Intestinal Epithelial Cells during Gut Mucosal Homeostasis and Regeneration

**DOI:** 10.3390/biomedicines12030668

**Published:** 2024-03-17

**Authors:** Hammed Ayansola, Edith J. Mayorga, Younggeon Jin

**Affiliations:** 1Department of Animal and Avian Sciences, University of Maryland, College Park, MD 20742, USA; ayansola@umd.edu (H.A.); jmayorga@iastate.edu (E.J.M.); 2Department of Animal Sciences, Iowa State University, Ames, IA 50011, USA

**Keywords:** intestinal stem-cell niche, mesenchymal stromal cells, subepithelial gradient factors, epithelial–mesenchymal interactions

## Abstract

Intestinal epithelial cell activities during homeostasis and regeneration are well described, but their potential interactions with stromal cells remain unresolved. Exploring the functions of these heterogeneous intestinal mesenchymal stromal cells (iMSCs) remains challenging. This difficulty is due to the lack of specific markers for most functionally homogenous subpopulations. In recent years, however, novel clustering techniques such as single-cell RNA sequencing (scRNA-seq), fluorescence-activated cell sorting (FACS), confocal microscope, and computational remodeling of intestinal anatomy have helped identify and characterize some specific iMSC subsets. These methods help researchers learn more about the localization and functions of iMSC populations during intestinal morphogenic and homeostatic conditions. Consequently, it is imperative to understand the cellular pathways that regulate their activation and how they interact with surrounding cellular components, particularly during intestinal epithelial regeneration after mucosal injury. This review provides insights into the spatial distribution and functions of identified iMSC subtypes. It focuses on their involvement in intestinal morphogenesis, homeostasis, and regeneration. We reviewed related signaling mechanisms implicated during epithelial and subepithelial stromal cell crosstalk. Future research should focus on elucidating the molecular intermediates of these regulatory pathways to open a new frontier for potential therapeutic targets that can alleviate intestinal mucosa-related injuries.

## 1. Introduction

The continuous self-renewal of the intestinal stem cells (ISCs) underpins the high turnover rate of differentiated epithelial cells [1,2,3,4]. Over a decade ago, Hans and colleagues demonstrated how ISCs self-renew and proliferate into progenitor cell populations before differentiating into specialized epithelium cells [5,6]. From their study, we understand that stem cell division is confined to the crypt. In contrast, proliferating and differentiated cells create a single-layered columnar epithelium. This layer consists of different cell types arranged from the crypt base toward the apical villi region [5,7,8]. The proliferating progenitor cells in the transit-amplifying (TA) zone undergo asymmetrical cell divisions, such that the half-daughter cells differentiate into specialized lineages, which include the secretory (Paneth, tuft, goblet, and enteroendocrine cells) and the absorptive enterocytes. They migrate and form the single epithelial layer of the villi compartment [9]. Comparatively, the crypt–villus axis is characterized by distinct cellular interactions and molecular signatures that maintain homeostasis [10,11,12]. For example, previous studies demonstrated that Paneth cells produce canonical Wnt ligands to maintain the stem-cell niche [13,14,15]. Likewise, stromal cells close to the sub-villi have been shown to communicate with the differentiated cells by providing bone morphogenetic proteins (BMPs) and non-canonical Wnt ligands [16,17,18,19,20].

During regeneration after mucosa injury, reserve stem cells are activated, and specialized cells dedifferentiate into active stem cell populations that proliferate to support epithelial regeneration in the damaged site [21]. Moreso, it was suggested that injured intestinal epithelial cells activate subepithelial fibroblasts via transforming growth factor-beta (TGF*β*) to support its proliferation and migration [22,23,24]. However, the specific roles of various subepithelial stromal cell populations in epithelial regeneration remain unresolved [25,26]. On this account, this paper reviews the recently characterized subpopulations of intestinal mesenchymal stromal cells (iMSCs) in mice and their interactions with the epithelium. It also presents a comprehensive overview of their roles in intestinal epithelial morphogenesis, homeostasis, and regeneration. To ensure the relevance of this review process, we explored Google scholar and NIH PubMed databases as the primary sources of the referenced materials. Eighty-two percent of the cited references span between 2013 and 2023. This fact reflects that this review was not only subjected to a rigorous process but also addressed relevant findings in recent years. Specifically, we searched keywords such as intestinal–epithelial and mesenchymal–cellular interactions, intestinal signaling pathways, and mesenchymal roles in inflammatory bowel diseases.

## 2. Intestinal Stem Cell Regulation

Neighboring epithelial and subepithelial cells secrete gradient factors that influence the ISC fate. These cellular interactions maintain ISC stemness at the base of the crypt and aid intestinal cell proliferation and differentiation activities along the crypt–villus axis [4,27]. Though the specificity of active stem cells is debatable, they are marked by LGR5^+^, OLFM4, ASCL2, RNF43, SOX9, MSI1, and SMOC2, because of their ability to self-renew and replenish differentiated epithelial cells [9,11,27,28]. Paneth cells that are adjacently interspersed between ISCs produce epidermal growth factor (EGF) and ligands (Wnt and Notch) to regulate ISC activities. For instance, Paneth cells as described by Sato, et al. [29] improved organoid formations. Yilmaz, et al. [30] also reported that caloric restrictions preserved ISC self-renewal, and their adaptation is likely coordinated by mTORC1 signaling in the neighboring Paneth cells. Other epithelial cell populations also help maintain intestinal epithelial balance by providing cues, which are highlighted in Table 1. Specifically, non-active stem cells, found in +4 positions, were reported to replenish the active stem cell pool during injury recovery through YAP1-dependent transient expansion [31]. Likewise, Paneth cells in the small intestine or *Reg4^+^-*expressing cells in the colon secrete canonical Wnt factors (such as *Wnt3*, *Wnt9b*, and *Wnt11*) and growth factors (EGF) to support epithelial regeneration via Notch signaling activation [14,32,33,34,35]. Despite their regulatory support to the ISC niche, debates remain on the contribution of these non-stem cell populations in the ISC niche. It was reported that Paneth cell depletion did not significantly alter ISC fate in vivo [36], and their functions can be substituted with exogenous Wnt supplements in ex vivo enteroid culture [37]. Additionally, recent studies have shown that iMSCs play indispensable roles in regulating intestinal morphogenesis, homeostasis, and regeneration by providing gradient factors to maintain intestinal integrity [38,39,40]. It is therefore vital to address how iMSCs regulate the Wnt, EGF, and BMP signaling pathways that confine active stem cell renewal at the crypt base and differentiate progenitors into specialized cells [2].

### 2.1. Signaling Pathways

#### 2.1.1. Wnt Pathway

ISC stemness, proliferation, and differentiation largely depend on ligand-mediated signaling processes. As demonstrated in previous studies, canonical Wnt (or Wnt/β-catenin) signaling is crucial for ISC self-renewal maintenance [44,45]. For example, knocking out *Porcupine* and *Wntless,* the essential mediators of Wnt secretion, greatly reduced the ISC population [18,43,46]. Similarly, the adenoviral delivery of the Wnt signaling antagonist, Dkk1, inhibited the intestinal proliferative marker, Ki67, and the Wnt/β-catenin target genes, CD44 and EpbB2 [47]. These studies confirmed the critical roles of Wnt signaling in intestinal homeostasis [2].

Notably, the canonical Wnt/β-catenin activation, regarded as a major promoter of ISC stemness maintenance, is triggered by ligands (e.g., Wnt2b and Wnt3) secreted from the ISC neighboring cells. First, the Porcupine protein synthesized in the endoplasmic reticulum of the neighboring cells initiates Wnt ligand secretions that bind on the frizzled receptors of the ISC plasma membrane and simultaneously attach to the LRP5/6 co-receptors [48]. This Wnt ligand–receptor binding initiates Dsh protein recruitment, leading to LRP receptor phosphorylation and the disruption of the Axin–GSK3–CK1 complex that orchestrates the β-catenin ubiquitination [2,11]. As a result, β-catenins are stabilized and accumulated in the cytoplasm. They subsequently translocate into the ISC’s nucleus to interact with LEF/TCF factors, thereby promoting canonical Wnt target gene transcription as illustrated in Figure 1. Blocking Wnt ligands may compromise Lgr5^+^ stemness and induce premature lineage differentiation [49]. This is partly due to the essential role of Wnt in ISC stemness maintenance. In contrast, the expressions of Wnt signal-related promoters such as Wnt2b and Rspondins (Rspos) and Wnt target genes increased the Lgr5^+^ population and ISC diminished as the proliferating cells migrated apically in the transit-amplifying region [38].

Unlike the canonical Wnt ligands that support intestinal stemness via β-catenin stabilization by orchestrating Wnt target gene transcriptions, non-canonical Wnt pathways are β-catenin-independent [50]. Non-canonical Wnt pathways regulate the Wnt/PCP (planar cell polarity)- and Wnt/Ca^2+^-dependent signaling cascades to modulate the ISC fate [51,52,53]. Despite the extensive knowledge surrounding canonical Wnt signaling, the function of non-canonical Wnt signaling has not been thoroughly studied. Non-canonical Wnt signaling, triggered by non-canonical Wnt ligands, promotes cellular differentiation and regeneration to facilitate ISC migration. Non-canonical Wnt ligands, including Wnt4, Wnt5, Wnt5b, and Wnt16, were remarkably expressed along the villus axis to the tip [20,38]. Their presence indicates that some Wnt ligands support villus differentiation and homeostasis, which might complement BMPs that promote intestinal maturation [20,54]. Recent scRNA sequencings have identified iMSC clusters that are abundant in the villus sub-region and express *Wnt4*, *Wnt5*, *Wnt 16*, and *Egf* [20,54]. This paradoxical revelation contradicts the classical roles of Wnt in maintaining the stemness [11,55]. For instance, ablation of villus tip subepithelial *Lgr5^+^* iMSCs, which expressed high *Wnt5a* and *Bmp4*, resulted in enterocyte loss [54,56]. Additionally, the Wnt/Ca2^+^ pathway can independently stimulate stem cell proliferation when Wnt5a/Fzd6 interactions increase intracellular Ca^2+^ levels in the cytosol during gastric cancer. This leads to PKC- and CAMK-mediated downstream cascades to promote cell migration [53,57,58]. In a related pattern, Wnt ligands could bind and activate Rho receptor complexes (Ror/Ryk) to trigger c-Jun N-terminal kinase, initiating JNK target gene transcription [57,59]. JNK target genes participate in cellular proliferation, migration, and regeneration by targeting exoskeleton proteins and adhesion molecules such as actin and Rho GTPases. These suggest that non-canonical Wnt signaling downstream could be a promising target for pharmaceutical drugs to improve intestinal injury recovery [60].

#### 2.1.2. BMP Pathways

The bone morphogenetic protein (BMP) pathway is an important signal opposing the Wnt mechanism that stimulates epithelial differentiation. Contrary to canonical Wnt signaling, BMP/Smad activities increase toward the apical region, indicating repressed stem cell activities along the villi [61,62]. An extensive review has addressed the BMP/Smad regulation of intestinal epithelial differentiation and homeostasis [62]. The focus here provides insight into how Bmp factors mediate the interaction between the intestinal epithelium and iMSCs. Of note, BMP4, a ligand stimulating Bmp signaling, was reported as a transcriptional target gene of hedgehog-responsive iMSCs [61,63,64]. The interaction between iMSCs and the epithelium is crucial for defining the distinct cellular characteristics of the crypt and villi. This interplay is achieved by supplying Wnt and Bmp gradients [65,66]. A recent study found that populations of PDGFRα^+^ iMSCs promote Wnt signaling, which can stimulate the expression of epithelial hedgehog ligands (Shh and Ihh) during morphogenesis [20]. In turn, epithelial-derived Shh ligands directly activate iMSC target gene transcriptions, including BMPs, that shape the developing intestinal villus formation [20,63,67]. Interestingly, subepithelial iMSC provides essential ligands that inhibit BMP activities in the crypt base where stem cell renewal and proliferation are at their peak. For instance, sub-cryptal iMSCs secrete Noggin and Gremlin1 to block BMP signaling, thereby supporting the ISC niche [20,68]. This means that epithelial–mesenchymal communications create a feedback loop to maintain intestinal homeostasis along the crypt–villus axis.

#### 2.1.3. Other Cellular Signaling Pathways

After active ISC proliferation, the fates of cell lineage specialization are intricately regulated by the Notch pathway. Notch signaling commits progenitor cells to transform into differentiated specialized cells [34,69]. Moreso, increased Notch ligands from neighboring Paneth cells redirect proliferating cells toward the enterocyte lineage by repressing *Math1* transcriptions [7,70]. The Paneth cell, which is a secretory lineage, could trigger negative feedback to limit secretory cell lineage commitment. These feedback mechanisms thereby support the importance of stringent regulations of ISC activities to maintain intestinal epithelial homeostasis. Notwithstanding the foregoing, other mechanistic pathways, such as EGF, contribute to ISC maintenance and are well documented in previous studies and reviews [2,11,27].

Another notable mechanism that regulates intestinal cellular interactions is the hedgehog (HH). HH ligands secreted by the epithelium initiate a negative regulation of its receptor (*Ptch1*) on neighboring iMSCs [64,71]. In the absence of HH ligands from epithelial cells, *Ptch1* inhibits Smoothened (Smo) signaling transduction. The Smo inhibition leads to Gli family phosphorylation in the cytoplasm by a degradation complex that includes glycogen synthase kinase 3 beta (GSK3β), casein kinase I alpha (CKIα), and protein kinase A (PKA). In contrast, when HH ligands bind on *Ptch1* receptors, they release Smo to activate STK36, which inhibits the degradation complex assembly. STK36 also phosphorylates the SUFU complex to stabilize Gli and allow nuclear accumulation, which subsequently stimulates Gli-dependent target gene transcriptions (Figure 2). Despite the extensive review of hedgehog regulations, especially their intricate functions in fetal villigenesis and regeneration [20,64,68], their roles in promoting crypt stemness during homeostatic conditions are just gaining attention in recent studies [45,65,72]. This was best described when hedgehog-responsive mesenchymal cells were revealed to constitute a key colonic stem-cell niche [72]. Consistently, the mesenchymal subsets that are localized in sub- and pericryptal regions form the main sources of the Wnt, Rspo, and Grem niche that supports the crypt stemness [38,39]. For example, the activated HH pathway upregulated stromal Wnt expression, which contributes to an increased OLFM4-positive stem cell pool [73]. Likewise, current data also suggest that HH/Wnt pathway crosstalk potentially promotes intestinal regeneration. Upon irradiation-induced intestinal epithelial injury, Shh was significantly upregulated, leading to increased production of Wnt ligands (Wnt2b, Wnt4, and Wnt5a) in the underlying stromal cells. Upregulated Wnt ligand production resulted in enhanced regeneration [74]. Together, these signaling networks complementarily or in opposing efforts regulate intestinal cell fate along the crypt–villi topologies during intestinal homeostasis and injury repair processes.

## 3. Intestinal Cell Plasticity and Regeneration

The intestinal epithelium is a delicate but resilient organ. It undergoes a rapid turnover of epithelial cells, predisposing it to luminal insults [75]. The exposure of epithelial cells to luminal contents can cause severe perturbations induced by physical or pathogenic agents [21]. These necessitate addressing the concept of intestinal cell plasticity. While active Lgr5^+^ stem cells continuously proliferate to replenish the epithelium during normal homeostatic conditions, the proliferating properties make them more susceptible to mucosal injuries, including irradiation or inflammation [76]. Studies showed that various mucosal injuries, such as inflammation, hypoxia, or irradiation, caused Lgr5^+^ stem cell loss, thereby exacerbating intestinal epithelium regeneration [77,78,79,80]. Though the Lgr5^+^ stem cell pool is depleted during intestinal injury [78], another stem cell population (known as reserve stem cells) is activated to replenish the active stem cell pool [35,81,82]. Following intestinal mucosal damage, these non-active Lgr5^−^ ISCs play a crucial role in regeneration and help maintain intestinal homeostasis [31,83,84,85]. For example, both DSS (Dextran sulfate sodium) and irradiation-induced Lgr5^+^ cell loss caused the proliferation of quiescent Clu^+^-expressing cells to reconstitute columnar base Lgr5^+^ cells and replenish the damaged epithelium during regeneration [31]. To buttress this idea, a previous study reported that the transitory loss of Lgr5^+^ stem cells did not disrupt the intestinal architecture, and this could result from the reserved stem cells recruited following Lgr5^+^ stem cell loss [82,83]. This hypothesis corroborates the activation of putative reserve stem cells, including Bmi1, Hopx, and mTert, during Lgr5^+^ cell depletion [82,83,86,87]. It is suggested that quiescent cells are less susceptible to mucosal injury and play critical functions during the regeneration process [82,83]. Reserve stem cell populations are less proliferative and express high anti-apoptotic and DNA repair genes [31,88]. Though these properties make them a great candidate to replenish the active stem cell pool after mucosal injury [89,90], researchers are still debating whether reserve stem cells at the +4 position of the crypt are the main driver of intestinal regeneration [28].

Others proposed that several differentiated epithelial cells undergo fetal-like reprogramming and revert to active Lgr5^+^ cells during intestinal regeneration [23,24,74]. Secretary lineage seems to be the most prominent among the differentiated cell types that support intestinal regeneration during mucosal healing [91]. The dedifferentiation by these non-putative active stem cells to replenish the Lgr5^+^ pool is known as intestinal plasticity [28,82,92]. In the last decades, lineage-tracing techniques using transgenic animals have provided the model to delineate cell migration [5,7]. This technique can track the specific cell lineage by editing the gene of interest. Van Es, et al. [42] explored this model to reveal that Dll1^+^, a secretory cell progenitor, can establish organoids containing Lgr5^+^ cells in in vitro culture. In a similar pattern, when irradiation depleted the stem cell pool in vivo, Dll1^+^ transformed into active Lgr5^+^ stem cells that proliferated into multiple progenitor lineages [42]. Another secretory progenitor cell, Atoh1^+^, was confirmed to generate Lgr5^+^ stem cells using different injury models such as irradiation and DTR (Diphtheria toxin receptor)-induced stem cell loss strategies [93]. To clarify the mechanisms behind the dedifferentiation phenomenon, several studies have tried to unravel the factors associated with cell plasticity and regeneration, but this area is still under active investigation. Recently, some studies suggested potential underlying mechanisms involved in the processes [21,28]. For instance, recombinant WNT3A supplements promote Dll1^+^ to produce organoids populated by Lgr5^+^ cells [94]. Furthermore, *Ascl2,* a Wnt target gene, was found to promote Paneth cell dedifferentiation following DSS treatment [84]. According to Yu, et al. [95], Notch signaling target genes, particularly *Hes1 and Notch1,* are upregulated during Paneth cell dedifferentiation in irradiation-induced damage studies. Likewise, *Yap1,* a key hippo signaling target gene, was significantly upregulated in a DSS-induced study during epithelial regeneration [96]. Altogether, the crosstalk among Wnt, Notch, hedgehog, and hippo signaling needs further studies. Future research will help us understand their contributions to epithelial regeneration [2,97]. Unraveling these mechanisms could provide key insights into therapeutic targets to alleviate mucosal healing.

Non-epithelial cells, such as immune and iMSCs, also contribute to intestinal cell plasticity and regeneration. These cells secrete factors, including growth factors and cytokines, to stimulate mechanistic signalings that regulate regenerative processes [43,98,99]. They sense and are recruited to respond to injury repair. Immune cells specifically induced inflammation and also recognized damaged-associated molecular patterns released by apoptotic cells [100]. In addition to immune cells, iMSCs are essential subepithelial cell types that are underappreciated during regeneration responses. Injured epithelial cells can send signals to iMSCs through hedgehog ligands, specifically Ihh and Shh, thereby enhancing ISC regeneration to support repair processes after epithelial injury [101,102,103]. Elevated levels of these ligands increased HH target gene transcription in iMSCs, including *Cyclin D1*, to promote epithelial regeneration [45,71,72]. Recent studies are unraveling potential iMSC subsets that are involved in intestinal epithelial repair functions. For instance, a study demonstrated that Gli^+^-expressing mesenchymal cells may secrete *Rspo3* to support epithelial repair processes using the DSS-induced damage model [72]. Moreover, DSS-induced colitis increased *fgf10*, *Vegf*, *Wnt2b*, *Grem1*, and *Rspo1* expressions in CD34^+^ cells. These studies buttressed iMSC roles in epithelial repair responses [45]. In conclusion, depleted active stem cell populations are certainly replenished during epithelial injury repair and regeneration, but there is still a knowledge gap on how iMSCs support stem cell regenerative capacity. Understanding intestinal epithelial and MSC interaction could be the solution to treating relapsing intestinal diseases, such as inflammatory bowel diseases and necrotizing enterocolitis. Thus, it is important to elicit the contribution of multicellular crosstalk during intestinal cell plasticity and regeneration programs. We will discuss more details on the role of iMSCs during regeneration in the next chapter.

## 4. Subepithelial Mesenchymal Stromal Cell Characteristics

Intestinal mesenchymal stromal cells (iMSCs) are among the cell population localized in the lamina propria. The term “iMSCs” represents a group of non-epithelial, non-endothelial, non-neuronal, and non-hematopoietic cells that contribute growth factors and chemokines to regulate the intestinal epithelial homeostasis, regeneration, and immune functions [104,105,106]. These cells share surface markers with mesenchymal cells found in other tissues, giving them similar characteristics [107,108]. They were previously categorized by non-expressing α-smooth muscle actin (α-SMA^−^) fibroblasts, α-SMA^+^-expressing myofibroblasts, pericytes, and mesenchymal stem cells [55,65,107]. Before recent evidence, iMSCs were largely focused on their structural support functions to the epithelial architecture. This is because they form the largest component of the extracellular matrix (ECM) of the intestine [106]. Contrary to these past views, recent studies using immunostaining and single-cell RNA sequencing have revealed iMSCs as heterogeneous cell populations capable of dynamic multidirectional crosstalk with epithelial, hematopoietic, and immune cells [39,43,109]. Accordingly, iMSCs are dispersedly distributed along the crypt–villi subepithelial region in the lamina propria and *muscularis mucosa* and deep in the *submucosa* with distinct morphology and functional support to epithelial cells.

### 4.1. Roles of Intestinal Mesenchymal Stromal Cells during Prenatal Intestinal Morphogenesis

The intestinal epithelium develops from endodermal cells in the embryo [67]. During the early fetal stage, these cells in tube-like sheet layers invaginate to form the pseudostratified epithelium (E9.5). This means that the proliferating epithelial cells are non-compartmentalized at the early embryonic stage [66,68,110]. Interestingly, cellular proliferation during early embryonic development survives in the absence of Wnt/β-catenin signaling despite the crucial role this mechanism plays in adult stem cell homeostasis [111]. Wnt target gene transcription factor (TCF) was found to be redundant during the pseudostratified epithelial development stage. Rather, mice lacking the *Tcf7l2* gene had significant epithelial shortening only after the villi had been formed completely [112]. In another study, conditional β-catenin deletion did not affect pseudostratified epithelium proliferation during the early fetal stage [113]. Together, these studies indicate that there might be distinct signaling mechanisms between early- and late-phase embryogenesis.

The pseudostratified intestinal epithelium and mesenchyme rapidly increase from E9.5 to E14.5. By E14.5, the pseudostratified epithelium undergoes extensive remodeling into a columnar epithelium, which leads to the emergence of a villus structure. To gain insights into the underlying factors that potentially shape fetal gut compartments at this stage, Maimets et al. reported that CD29 and PDGFRα^+^ cells are crucial for fetal gut vilification [20,46,68,111,114]. The CD29^+^ cells expressed high *Acta2, Myl4, Des,* and LRIG1 levels, which could be the progenitor for fetal gut *muscularis mucosa* cells. More importantly, they revealed how PDGFRα^+^cells guide the villigenesis of the pseudostratified epithelium [20,66]. This suggests that PDGFRα^+^cells could be a progenitor for stromal lineages because of their indispensable contributions to villi formation during morphogenesis, which aligns with previous studies that demonstrated iMSCs drive intestinal epithelial fate [20,65,115,116]. They hypothesized that PDGFRα^+^cells are crucial for villi emergence because they form clusters close to the expanding epithelium. This view is supported by their in vitro model study, which showed that PDGFRα^+^ cells isolated from the fetal gut can promote organoid growth independent of essential growth factor supplements. In contrast to the early fetal stage, they also found that inhibiting *Porcn* in pregnant dams (a Wnt ligand upstream regulator) between E12.5 and E16.5 had no effects on the fetal colon, but abrogated SI villi formation. Based on these data, we could deduce that Wnt gradients become relevant from mid-stage morphogenesis onward for maintaining the crypt–villus compartment. To corroborate this hypothesis, other studies have also highlighted that the underlying mechanisms of early embryonic morphogenesis are biologically different from those that regulate the postnatal intestinal epithelium [113,117].

Embryonic mice at E14.5 exhibit rapidly expanding epithelial cells. By E16.5, these proliferating cells are restricted to the intervillous domain between neighboring villi and differentiated absorptive and secretory cells in villi [67]. A previous report suggests that embryonic iMSCs expressing *Dlk1* support the rapid expansion of fetal gut morphogenesis, which differs from their role in adult mice [117]. To confirm this, PDGFRα^+^ cells, which are the dominant iMSC cluster in the late fetal phase, reportedly expressed DLK1, but this expression decreased toward late gestation [117]. For these reasons, the current challenge is to clarify the functional diversities of the PDGFRα+ subset as recent studies classified them as PDGFRα^hi^, PDGFRα^medium^, and PDGFRα^lo^ cells during morphogenesis [20,44]. These subsets showed distinct characteristics and potentially have unique signaling properties.

Studies have previously predicted the indispensable roles of hedgehog signaling, involving Shh upregulation during villi formation [20,66,67]. Consistently, hedgehog target genes, especially *Ptch1*, were detected in the absence of Wnt signaling. *Ptch1* is a key receptor, expressed by mesenchyme, for the Shh ligand, indicating that mesenchymal cells are indispensable during fetal gut formation. While the roles of non-Wnt pathways in fetal intestinal patterning during embryogenesis remain unclear, the available data suggest multiple instances of signaling crosstalk, such as Wnt, hedgehog, and Notch, among other factors produced by epithelial–mesenchymal cells that regulate fetal gut morphogenesis [20,113].

### 4.2. Roles of Intestinal Mesenchymal Stromal Cells during Intestinal Homeostasis

The neonate and mature intestinal epithelium are supported by different stromal cell populations [38,39,44]. By the time of birth, the neonate gut will transition into a compartmentalized intestine driven by polarized signaling mechanisms. Specifically, the crypt and Paneth cells in the small intestine will start to emerge by the 14th postnatal day. By postnatal day 28, crypts will have rapidly expanded and matured, forming the intestinal epithelial crypt–villus structure [67,118,119]. Adult mice maintain intestinal homeostasis through a range of cellular programming, including ISC stemness, TA proliferation, and differentiation [9]. Each process requires unique gradient factors to keep the intestinal compartments in normal condition, in which iMSCs serve as a crucial gradient source for all the essential ligands required for the regional specificity of epithelial cell integrity. Remarkably, iMSC subtype heterogeneity increases as gut maturation progresses during postnatal development [39,44].

#### Recent Classification of iMSC Lineage

Although there has been substantial progress in characterizing epithelial cells [21], subepithelial iMSC classifications and their cellular diversity and functionality remain challenging due to a lack of unique surface markers. Nevertheless, they are broadly categorized according to their regional specificity and functional diversity along the intestinal region using immunostaining, FACS, and transgenic mouse models. iMSCs expressing PDGFRα^hi^, ACTA2^lo^ myofibroblast, FOXL1^+^, and GLI1^+^ but not vascular CD31^+^ markers are loosely regarded as telocytes due to their proximity to epithelial cells in the villus region [68,120,121]. Most cryptal iMSCs uniquely express CD34^+^/GP38^+^ co-localizing with CD81^+^ and other PDGFRα^low^ cells, forming sub- and pericryptal stromal populations [44,45,46,68]. The subcryptal stromal populations produce trophic factors to support the intestinal stem-cell niche, and they are distinguished from *muscularis mucosa* cells due to their lack of *Myh11^+^* gene expression [43]. Despite being isolated and studied, those cells still display heterogeneity and remain undifferentiated in terms of their origin and possible lineages.

Recent studies using scRNA-seq assay have revealed a robust description of possible lineage and anatomic and physiologic heterogeneity of previously obscure iMSC subtypes [38,39,43]. One common proposition is that adult iMSC subsets originate from a similar embryonic precursor identified as Gli-expressing cells, which is still debatable [39]. Likewise, most adult iMSCs express PDGFRα^+^ at varying magnitudes, which suggests PDGFRα^+^ as another potential progenitor—more details on how iMSCs support embryogenesis are presented in the morphogenesis section [20,66]. Their localization probing and gene expression profiling showed a diverse population of iMSCs despite sharing the same lineage.

iMSC subsets that are sparsely localized beneath the epithelium produce gradient factors that shape the crypt/villi compartments. To appreciate the spatial distributions of these subsets, we reviewed recently characterized iMSCs and schematically illustrated them in Figure 3. These studies revealed distinct iMSC subsets that are classed under PDGFRα^lo^ (CD34^hi^CD81^+^, CD34^hi^Igfbp5^+^, and CD34^lo^Fgfr2^+^) and PDGFRα^hi^ (CD9^hi^CD141^-^, CD9^lo^CD141^+^, and CD141^int^) [39].

##### Functions of Pericryptal (PDGFRα^lo^ or CD34^+^Gp38^+^) Subpopulations

PDGFRα^lo^ subsets are found beneath the *muscularis mucosae* (CD81^+^), pericryptal (Igfbp5^+^), and the lamina propria (Fgfr2^+^). This idea is strengthened by the data reported in recent studies [38,39]. The studies suggested that gradients regulating the ISC niche and terminal differentiation support the intestinal epithelial regional specificity.

To confirm the functional characteristics of these PDGFRα^lo^ subsets, clusters of PDGFRα^lo^ cells that expressed pericryptal CD34^+^Gp38^+^ only emerged after birth and they are found to promote intestinal stemness [45]. The study demonstrated that CD34^+^Gp38^+^ cells are the main Wnt ligand contributors to adult ISC homeostasis when the crypt is formed. This indicates that the intestinal epithelium requires a specific iMSC subpopulation to reach maturation postnatally [45]. The idea corresponds with data showing that CD81^+^ cells located in the *muscularis mucosae*, a subset of the CD34^+^ population, are the key producer of *Grem1* that inhibits Bmp activities in the crypt [39,45]. PDGFRα^lo^CD81^+^ *Ackr4^+^* (trophocytes) and PDGFRα^lo^CD81^-^CD55^hi^ cells are subsets of the CD34^+^ population. They produce BMP antagonists for the stem-cell niche to promote ISC stemness near the crypt base [38]. The Shivdasani group also reported that PDGFRα^lo^CD81^+^ cells expressed high *Wnt2b*, *R-spos*, and *Grem1* levels [55,65]. The depletion of *Rspo3-*secreting iMSCs that caused delayed gut maturation and reduced ISCs buttresses the functional specificity of PDGFRα^lo^CD81^+^ cells [46,49,122].

The cryptal cells transitioned into differentiated cells just above the villus base. iMSCs that are found beneath the villus base and the corresponding lamina propria region produce pro-BMP factors [38,39]. For example, PDGFRα^lo^CD81^-^CD55^lo^ and CD34^lo^Fgfr2^+^ cells expressed non-canonical *Wnt4* near the top of the colon crypt base, suppressed trophic factor effects, and also reinforced the BMP gradient activities. These functions suggest their roles in promoting terminal differentiation in the TA domain toward the villus region. Though they are PDGFRα^lo^ subsets, their transcriptomic data revealed they expressed a low level of *Wnt2b*. This is consistent with previous studies that showed BMP gradients increased apically in the SI villi or the colon crypt top [38,65]. In contrast, BMP inhibitors increased distally, especially from pericryptal PDGFRα^lo^CD81^−^ cells to PDGFRα^lo^CD81^+^ trophocytes beneath the *muscularis mucosae* [39,44,45]. Though it is currently difficult to sort, culture, and investigate all of these distinct iMSC subpopulations, studies have shown that they may functionally overlap, express related molecular profiles to perform complementary roles, and, in some cases, provide opposing cues to maintain intestinal homeostasis and development (Table 2). Collectively, the current observations depict that epithelial development relies heavily on gradients from diverse iMSCs to maintain intestinal integrity and the high turnover rate of epithelial cells [104,123,124].

##### Functions of PDGFRα^hi^ Subpopulations

PDGFRα^hi^Foxl1*^+^* cells are reported to be a good source of BMP-promoting factors, including *Bmp4*, *Bmp5*, and *Bmp7* [38,39,65]. Paerregaard et al. reported that the three PDGFRα^hi^ subsets expressed *Foxl1*. (1) *Nrg1* was expressed in CD9^hi^CD141^−^ cells; (2) *Cxcl12* and *Acta2* were expressed in CD9^lo^CD141^+^ cells; and (3) *Adamdec1*, *Wnt4*, and *Acta2^+^* were expressed in CD141^int^ cells [39]. While all these PDGFRα^hi^ subpopulations produce BMP (*Bmp5* and *Bmp7*) gradients, CD9^lo^CD141^+^ had the highest levels of *Wif1*, *Bmp3*, and *Bmp4*, contributing to terminal differentiation of intestinal epithelial progenitor cells. Upregulation of Wif1, a protein that binds and inhibits canonical and non-canonical Wnt ligands (such as Wnt3a, Wnt4, and Wnt5a) [126], corroborated the hypothesis that Wnt signaling activities are repressed from the villus base toward the tip. Interestingly, using RNA velocity analysis, these three PDGFRα^hi^ subsets were predicted to originate from CD34^lo^Fgfr2^+^ found along the lamina propria of the small intestine in mice.

According to a recent study, the postnatal intestinal epithelium required PDGFRα^+^ cell-dependent maturation to shift from immature proliferative compartments into distinctly functional crypt–villi regions [44]. In their study, they generated reporter and inducible lineage-tracing models for lymphotoxin beta receptor (LTβR) cells by crossing LTβR*^tTA^* mice with Rosa*26^floxSTOP-YFP^* mice. Using the inducible lineage-tracing technique, some fractions of PDGFRα^+^ subepithelial stromal cells developed from the LTβR^YFP^ progenitor cell lineage before the mice reached weaning age. To confirm this hypothesis, they also generated a direct LTβR^GFP^ reporter model to show that PDGFRα^hi^ cells, found close to the villus epithelial cells, expressed GFP [44]. Additional transcriptomic analysis revealed that LTβR^+^PDGFRα^+^-expressing cells showed upregulated levels of hedgehog and pro-differentiation gene markers that are specific to the PDGFRα^hi^-expressing cell population, especially *Ptch1*, *Foxf2*, *Gli1*, *Bmp4*, and *Bmp5*. In contrast, the pro-stem-cell niche gene markers (i.e., *Cd34*, *Cd81*, *Rspo2/3*, and *Grem1/*2) were downregulated in the transcriptome data of LTβR^+^PDGFRα^+^ cells [44,65]. Inducible conditional ablation of PDGFRα in the LTβR lineage (LTβR^ΔPDGFRα^ mice) caused an increase in ISC markers (*Olfm4^+^* and *Lgr5^+^*) in the gut compared with the wild-type. Consistently, the LTβR stromal lineage from LTβR^ΔPDGFRα^ mice had a decreased pro-differentiating factor (specifically, *Bmp2*). Pro-stemness factor (*Grem2* and *Chrdl1*) levels are increased in their stromal cell transcriptome data. Collectively, these observations demonstrated in their studies suggest that the LTβR^+^PDGFRα^+^ cells expressing pro-differentiation signals share similar characteristics with the PDGFRα^hi^ subsets reported by Paerregaard et al. [39,44,45]. These findings indicate that they are essential for intestinal epithelial maturation before weaning. More importantly, their results showed that the appearance of PDGFRα^hi^ cells not only influences epithelial cell differentiations but is also essential for the transcriptomic switch of functionally distinct LTβR^+^ stromal lineage cells toward maturation and localization during early postnatal gut development [44]. Accordingly, different iMSC subsets create a functionally distinct enabling environment for polarized signaling crosstalk to regulate epithelial cell fate during development. To sum it up, iMSC populations are less diverse during intestinal morphogenesis but become heterogeneous postnatally to support mature intestinal epithelial homeostasis [38,39,43,68].

### 4.3. Roles of Intestinal Mesenchymal Stromal Cells during Intestinal Injury and Repair

iMSCs are not only crucial for intestinal morphogenesis and homeostasis; recent reports have also revealed their modulatory functions using DSS, irradiation, and DTR-induced injuries in transgenic mice. For example, RBP1^+^ cells, confirmed to be a subset of GLI1^+^ cells, are suggested to sense DSS-induced colitis injury, thereby stimulating *Rspo3* production to promote the injury recovery process [72]. Likewise, the conditional deletion of the *Wntless* allele in a *Villin*-Wls^cKO^, a protein required for Wnt secretion, triggered GLI1^+^ cell expansion during injury. This observation was confirmed using recombinant Shh ligand to promote GLI1^+^ cell response to injury by compensating for epithelial Wnt loss. The results suggested that Shh signaling could be a potential regulatory target for enhancing mucosal healing by activating iMSCs. In another study, pericryptal CD34^+^ cells, in response to DSS treatment, reportedly migrate and localize under the colonic crypt–apical epithelial cells [127]. Following the induced epithelial injury, the mRNA data revealed upregulated levels of *Bmp2*, *Bmp3*, *Bmp7*, and *Wnt5a*, indicating the plasticity potential of CD34^+^ to support reepithelization during colon regeneration [127].

According to the reviewed studies above, mesenchymal cells support epithelial regeneration by secreting growth factors and other signaling molecules that promote the proliferation and differentiation of epithelial cells. scRNA seq analyses showed that iMSCs are intricately involved in epithelial cell responses to injury recovery processes. iMSCs such as the CD81^+^ and CD81^−^ subsets play significant roles during injury states, by increasing the production of Wnt-promoting factors [45,72]. For instance, *Grem1* and *Rspo3* reportedly increased PDGFRα^lo^ cells during colitis-induced conditions. The data suggested that these iMSCs support epithelial regeneration [116]. In another study, non-coding RNAs (miR-143/145) specific to iMSCs were also involved in regulating IGF1 signaling response to repair DSS-induced epithelial damage [128]. In all, iMSCs are essential for the repair and regeneration of the intestinal epithelium following injury. However, more studies are needed to clarify the relevance of these specific iMSC subsets during repair processes.

## 5. Conclusions and Future Perspectives

In this review, we discussed the spatial distribution of iMSC subtypes and their intricate relationship with epithelial cells. These cellular interactions play pivotal roles during intestinal morphogenesis, homeostasis, and regeneration. By providing signaling cues, different iMSC subsets have distinct regulatory functions along the crypt–villi axis. We know that iMSCs, such as GLI1^+^ and CD34^+^ populations, are recruited to increase the production of Rspodin and Gremlin gradient factors during experimental colitis models. However, current knowledge has yet to establish the specific homogenous iMSC subsets that promote improved mucosal injury repair, which could help achieve lasting remission in IBD patients. To bridge this research gap, our comprehensive review highlights iMSC pro-regenerative functions and the implicated mechanistic pathways. The enhanced gradient factor secretions by iMSC during regeneration suggest that they can be primed and used as therapeutic agents to promote regeneration. On this account, this review provides an overview of recent progress on iMSC characterizations. It also presents their potential relevance in the emerging role of cell-based therapy—a promising clinical approach to treat recursive inflammatory diseases, such as IBD. To achieve this breakthrough, we must first address the existing challenges facing iMSC classifications. This will allow us to understand better the characteristics, functions, and responses of each homogenous subset during homeostasis and epithelial regeneration. Consequently, three critical areas demand immediate attention to advance our understanding of iMSCs.

Firstly, researchers need to develop specific markers to further characterize homogenous iMSC subsets within clusters of notable iMSC subpopulations addressed in this paper. This approach will help to address the conflicting results on the iMSC population that have been reported to provide opposing signaling factors they secrete at distinct locations along the crypt–villi axis. Improved iMSC sub-type sorting techniques would clarify the unique functional diversity of specific homogenous iMSCs and how each distinct subset interacts with different epithelial cells. As such, researchers will be able to define the contributions of the overlapping molecular signatures by the heterogeneous iMSC clusters during normal and regeneration conditions. Thus, this is a call to develop novel iMSC surface markers for cell-sorting assays, which will advance the current knowledge about epithelial and mesenchymal crosstalk.

Second, to unravel the roles of characterized iMSC subsets during morphogenesis and disease development, researchers should improve lineage-tracing techniques that focus on examining the response of iMSCs to regeneration. For example, iMSCs are currently proposed to have multidirectional relationships with both epithelial and immune cells. While iMSCs supply Wnt and Bmp agonists during normal conditions, it remained obscure whether (1) they directly provide regenerative factors to repair epithelial cells, (2) adopt homing effects to replace damaged epithelial cells, or (3) indirectly activate anti-inflammatory immune cell responses to restore intestinal homeostasis after injury. Future research directions in this field could include investigating the mechanisms by which subepithelial stromal cells regulate stem cell behavior and the role of these cells in chronic diseases of the intestine, such as inflammatory bowel disease. Additionally, understanding the interactions between subepithelial stromal cells and other cell types may lead to the development of novel therapeutic strategies for promoting intestinal regeneration and repair.

Finally, despite the difficulty relating to investigating functional studies in vivo due to their delicate nature, the future approach needs to improve scRNA seq techniques, develop special iMSC culture conditions that can support the viability of FACS cells, and improve co-culture models to better understand the complexity of mesenchymal and epithelial crosstalk.

In conclusion, this paper has provided a comprehensive review of the recent evidence about iMSC heterogeneity. This diversity enables iMSCs to perform distinct or overlapping functions in maintaining intestinal epithelial integrity. Future studies should reconstruct 3D organoid co-culture set-ups, such as transwell and scaffold models, to elucidate the spatial organization of iMSCs in the intestine for different developmental stages. The research designs can reveal potential communications between epithelial and sorted iMSC subsets. By addressing these challenges, we can gain a better understanding of the complexity of the iMSC niche and develop novel therapeutic strategies for promoting intestinal regeneration and repair.

## Figures and Tables

**Figure 1 biomedicines-12-00668-f001:**
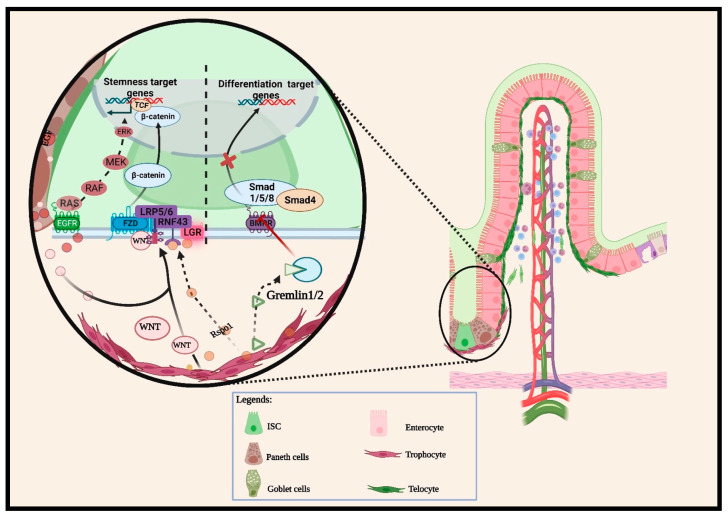
Multiple signaling pathways regulating intestinal epithelial–mesenchymal crosstalk in the crypt. The diagram provides an overview of the intricate mechanisms involved in the interactions between crypt-based epithelial cells and the neighboring iMSCs. This interplay controls intestinal stem cell (ISC) homeostasis and epithelial differentiation. The activation of Wnt-promoting pathways and inhibition of Bmp/Bmpr binding orchestrate ISC stemness in the crypt base. Paneth cells secrete Wnt, Notch, and EGF ligands that induce Wnt target gene transcription in ISC. Wnt ligands secreted by subcryptal iMSCs bind FZD and LRPs co-receptors. The ligand–receptor binding stabilized β-catenin in the Wnt signaling cascade to promote ISC-related gene transcription. Rspos binding LGR family receptors stabilized FZD expressions, contributing to WNT pathway activation. In addition, subcryptal iMSCs secrete Bmp antagonists such as *Gremlin* to maintain Wnt activities in the ISC niche.

**Figure 2 biomedicines-12-00668-f002:**
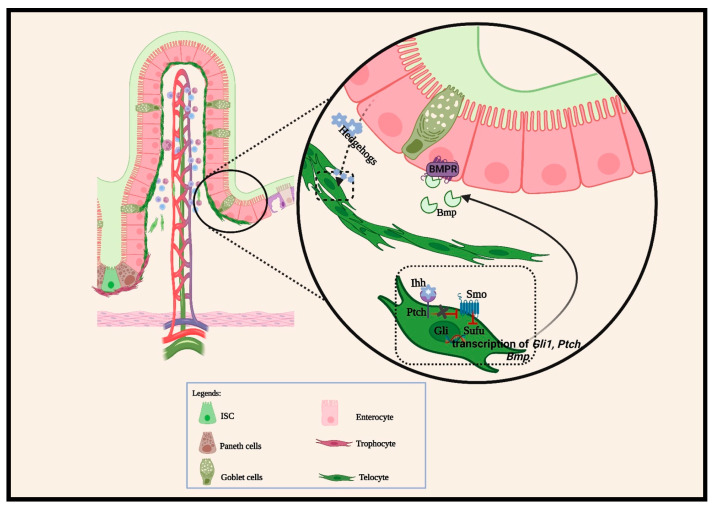
Intestinal villus base epithelial-secreted ligands activate hedgehog signaling in neighboring mesenchymal cells. These epithelial–mesenchymal interactions at the villus base stimulate intestinal epithelial cell terminal differentiation and coordinate the migration of differentiated cells toward the villus tip. Hedgehog ligands produced by epithelial cells bind on the *Pitch1* receptor of PDGFRα^hi^ cells located at the villus base. This binding triggers the transcription of hedgehog target genes. The genes transcribed by the Gli family of PDGFRα^hi^ cells include *Bmps.* Secreted Bmp ligands from these iMSCs bind with the Bmp receptor (Bmpr) on adjacent epithelial cells. The Bmp and Bmpr binding subsequently phosphorylate the Smad family transcription factor, which in turn induces the differentiation of villus epithelial cells.

**Figure 3 biomedicines-12-00668-f003:**
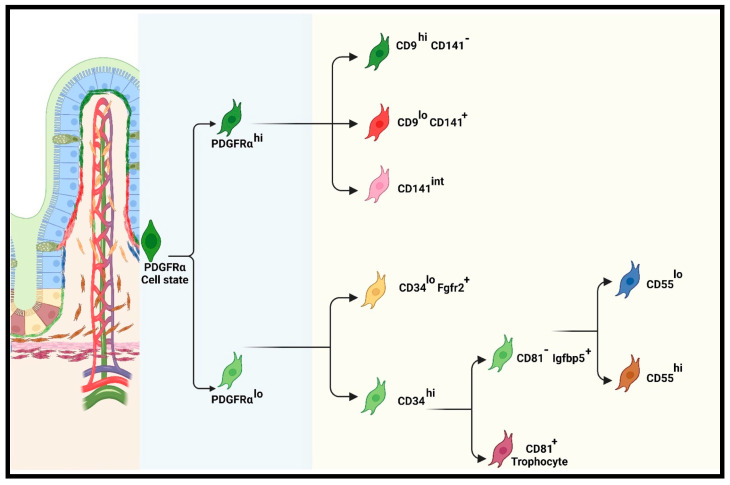
Schematic representation of intestinal mesenchymal stromal cell subsets and their location. The illustration here describes the subpopulations of PDGFRα-expressing cells, including telocytes, CD81^−^ cells, and trophocytes. The trophocyte cluster group, CD81^+^, is confined to the subcryptal domain, beneath the *muscularis mucosa*, to secrete Wnt-promoting factors that support the ISC niche. Other PDGFRα^lo^ subsets, including CD55^hi^ and Fgfr2^+^ cells, localize in the lamina propria and extend upward to the TA domain/villi trunk to initiate terminal differentiation. They switch the signal gradients from Wnt-promoting factors to Bmp agonists. PDGFRα^hi^ subsets form the subepithelial stromal populations that are localized in the villi core in the small intestine and the colon top [38,39].

**Table 1 biomedicines-12-00668-t001:** Key epithelial cellular players maintaining intestinal balance.

Epithelial Cells	Cell Markers	Ligands	Functions	Signaling Pathways	References
**Paneth cells**	*Defa4-*expressing cell	*Wnt3a*, *Wnt9b*, *Wnt11*	Support regeneration	Notch	[33]
	pS6^+^	*Notum*	Wnt inhibitor	Non-canonical Wnt/mTORC1	[41]
**Progenitor cells**	Dll^+^	*Dll1*, *dll4*	Support regeneration	Notch	[42]
**Secretory lineage**	Unknown	*Egf*, *Tgfa*,	Promote IEC homeostasis	EGFR/RAS	[32]
**Colonic Paneth cells**	REG4^+^	*Dll1, Egf*	Promote stemness	Notch	[43]
**+4 quiescent cells**	BMI1, HOPX, MTERT	Unknown	Support regeneration	Hippo	[21,31]
**Tuft cells**	DCLK1	*Dll1*	Promote regeneration	Notch	[35]

**Table 2 biomedicines-12-00668-t002:** Distinct intestinal stromal subsets supporting intestinal balance regulations.

Non-Epithelial Cells	Cell Markers	Ligands	Functions	Signaling Pathways	References
**PDGFRα^lo^ Cells**	CD81^+^, CD55^hi^	*Wnt2b*, *Gremlin1/2*, *Rspo3*	Wnt promoters	Wnt/β-catenin	[38,65]
	Fgfr2^+^, CD55^lo^	*Wnt4*, *Frzb*, *Sfrp1*	Wnt repressors	Non-canonical Wnt/Bmp	[38,39]
**PDGFRα^hi^ Cells**	FOXL1^+^, CD9^hi^CD141^−^, CD9^lo^CD141^+^, CD141^int^	*Bmp3/4*, *Wnt5a/b*, *Dkk*	BMP agonists, Wnt inhibitor	Non-canonical Wnt/Bmp	[39]
**PDGFRα^+^**	PDGFRα^+^DLK1^+^	*Dlk1*	Embryonic morphogenesis	Notch	[117]
**LTβR^+^**	LTβR^+^PDGFRα^hi^	*Pdgf*	Stromal maturation	Bmp activation	[44]
**Smooth muscle cells**	*Tagln^+^*, *Acta2^+^*, *Myh11^+^*	*Wnts*	ISC integrity and wound healing	Wnt/β-catenin	[43]
**Immune cells**	ILC2, ILC3	*Il13*, *Il22*	Promote regeneration	Wnt/β-catenin	[125]

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
