# Peer review of "Subepithelial Stromal Cells: Their Roles and Interactions with Intestinal Epithelial Cells during Gut Mucosal Homeostasis and Regeneration"

_biomedicines, 2024, doi:10.3390/biomedicines12030668_

Round 1

Reviewer 1 Report

Comments and Suggestions for Authors

The review is clear, comprehensive and relevant to the field. Thoroughly discusses the role and interactions of subepithelial stem cells with intestinal epithelial cells during intestinal mucosal homeostasis and regeneration. The use of state-of-the-art methodologies, such as single cell RNA sequencing to identify specific subgroups of intestinal mesenchymal stem cells (iMSCs), adds depth and novelty to the study. The review identified a significant knowledge gap regarding the specific functions of different subepithelial stem cell populations in epithelial regeneration. This gap represents an opportunity for future research to elucidate molecular intermediary regulatory pathways, potentially opening up new therapeutic opportunities. The inclusion of new clustering techniques and recent discoveries suggests that the review provides an up-to-date insight into the field.The review contains a significant number of recent and important references, with a significant number of citations in the last five years. This indicates a balance between grounding the review in well-established research and the inclusion of recent findings. The citations appear to be relevant to the statements made. 

The statements and conclusions drawn are consistent and well supported by the cited sources. The review systematically discusses the role of iMSCs in intestinal morphogenesis, homeostasis and regeneration, linking these processes to specific cellular interactions and signalling mechanisms.

The document contains excellent drawings and tables that illustrate the concepts discussed, although a detailed evaluation of each concept is not provided here. Typically, such visual aids in scientific reviews are intended to supplement the text, making complex data easier to understand and interpret.

In summary, the review appears to meet the criteria for a valuable scientific contribution, addressing the knowledge gap with recent and relevant references and providing coherent arguments supported by cited literature. 

Author Response

Dear reviewer, 

Thank you for giving us the chance to address the point raised by the reviewers and submit a revised draft of the manuscript “Subepithelial stromal cells: their roles and interactions with intestinal epithelial cells during gut mucosal homeostasis and regeneration” for publication in the Biomedicines journal. Your insightful comments and those of the reviewers are used to improve the quality of the revised draft. The affected changes are highlighted in yellow. Kindly see below, in red color, for a point-by-point response to the comments and concerns of the reviewers. Also, the page number corresponds to where the changes are made in the revised manuscript.

1: The review is clear, comprehensive and relevant to the field. Thoroughly discusses the role and interactions of subepithelial stem cells with intestinal epithelial cells during intestinal mucosal homeostasis and regeneration. The use of state-of-the-art methodologies, such as single cell RNA sequencing to identify specific subgroups of intestinal mesenchymal stem cells (iMSCs), adds depth and novelty to the study. The review identified a significant knowledge gap regarding the specific functions of different subepithelial stem cell populations in epithelial regeneration. This gap represents an opportunity for future research to elucidate molecular intermediary regulatory pathways, potentially opening up new therapeutic opportunities. The inclusion of new clustering techniques and recent discoveries suggests that the review provides an up-to-date insight into the field. The review contains a significant number of recent and important references, with a significant number of citations in the last five years. This indicates a balance between grounding the review in well-established research and the inclusion of recent findings. The citations appear to be relevant to the statements made. 

Author response: Thank you for providing a thorough review.

2: The statements and conclusions drawn are consistent and well supported by the cited sources. The review systematically discusses the role of iMSCs in intestinal morphogenesis, homeostasis and regeneration, linking these processes to specific cellular interactions and signaling mechanisms.

Author response: Thank you! Your feedback highlights the relevance of the work.

3: The document contains excellent drawings and tables that illustrate the concepts discussed, although a detailed evaluation of each concept is not provided here. Typically, such visual aids in scientific reviews are intended to supplement the text, making complex data easier to understand and interpret.

Thank you for pointing out the need to provide a detailed evaluation of schematic diagrams presented in the manuscript. While this information is summarized as legends to each figure/table, we addressed and provided comprehensive insights where the figures/tables are cited in the corresponding (sub)sections of the manuscript.

In summary, the review appears to meet the criteria for a valuable scientific contribution, addressing the knowledge gap with recent and relevant references and providing coherent arguments supported by cited literature. 

Author response: Thank you for the overall feedback. We appreciate your time and effort used to review our manuscript.

Reviewer 2 Report

Comments and Suggestions for Authors

This review provides a valuable resource for researchers interested in understanding the complex interactions between multiple types of cells in the intestinal epithelium and  iMSCs. The following are the comments provided for this paper.:

  1. The title of Table 1 is absent.
  2. In fact, this paper elucidates the signal association between iMSCs and diverse cell types of intestinal epithelium. However, since only intestinal epithelium cells are mentioned in the title. Thus, it is necessary to revise the title.
  3. Since the ultimate goal is to identify potential therapeutic targets for intestinal mucosa-related injuries, the review could explore the current and future applications of iMSCs in clinical settings. This could include discussions on the translational potential of these findings and the challenges associated with their implementation in patient care.
  4. The paper concludes by mentioning future research directions, but a more comprehensive outlook on the subject would be valuable. Suggested areas for further exploration could include new technological advancements that may enable a more comprehensive understanding of iMSCs, as well as the identification of new therapeutic targets based on the molecular intermediates discussed.
Comments on the Quality of English Language

Minor editing of English language required

Author Response

Dear reviewer,

Thank you for giving us the chance to address the point raised by the reviewers and submit a revised draft of the manuscript “Subepithelial stromal cells: their roles and interactions with intestinal epithelial cells during gut mucosal homeostasis and regeneration” for publication in the Biomedicines journal. Your insightful comments and those of the reviewers are used to improve the quality of the revised draft. The affected changes are highlighted in yellow. Kindly see below, in red color, for a point-by-point response to the comments and concerns of the reviewers. Also, the page number corresponds to where the changes are made in the revised manuscript.

This review provides a valuable resource for researchers interested in understanding the complex interactions between multiple types of cells in the intestinal epithelium and iMSCs. The following are the comments provided for this paper.:

1. The title of Table 1 is absent.

Author response: Thank you for pointing out the omission. As suggested, the Table 1 title has been updated in the revised manuscript.

2. In fact, this paper elucidates the signal association between iMSCs and diverse cell types of the intestinal epithelium. However, since only intestinal epithelium cells are mentioned in the title. Thus, it is necessary to revise the title.

Author response: Thank you for your suggestion. However, the current title includes the most important two cell types, iMSCs and epithelium. We believe that adding all the different cell types in the title would make it too long. Therefore, we think that the current title is concise enough to introduce the topics that are reviewed in the manuscript.

3. Since the ultimate goal is to identify potential therapeutic targets for intestinal mucosa-related injuries, the review could explore the current and future applications of iMSCs in clinical settings. This could include discussions on the translational potential of these findings and the challenges associated with their implementation in patient care.

Author response: Thank you for your excellent suggestion. As highlighted, in yellow, from line 537 in the revised manuscript, we have shared our perspective on the current challenges and future directions related to the potential therapeutic functions of iMSC during intestinal epithelial regeneration – “By providing signaling cues, different iMSC subsets have distinct regulatory functions along the crypt-villi axis. We know that iMSCs, such as GLI1+ and CD34+ populations, are recruited to increase the production of Rspodin and Gremlin gradient factors during experimental colitis models. However, current knowledge is yet to establish the specific homogenous iMSC subsets that promote improved mucosal injury repair, which could help achieve lasting remission in IBD patients. To bridge this research gap, our com-prehensive review highlights iMSC pro-regenerative functions and the implicated mechanistic pathways. The enhanced gradient factor secretions by iMSC during regen-erations suggest that they can be primed and used as therapeutic agents to promote regeneration. On this account, this review provides an overview of recent progress on iMSC characterizations. It also presents their potential relevance in the emerging role of cell-based therapy – a promising clinical approach to treat recursive inflammatory dis-eases, such as IBD. To achieve this breakthrough, we must first address the existing challenges facing iMSC classifications. This will allow us to understand better the char-acteristics, functions, and responses of each homogenous subset during homeostasis and epithelial regeneration. Consequently, three critical areas demand immediate attention to advance our understanding of iMSCs.”

4. The paper concludes by mentioning future research directions, but a more comprehensive outlook on the subject would be valuable. Suggested areas for further exploration could include new technological advancements that may enable a more comprehensive understanding of iMSCs, as well as the identification of new therapeutic targets based on the molecular intermediates discussed.

Author response: Thank you for your feedback. From line 537 in the manuscript, we presented 3 critical areas that will advance our understanding of iMSC identity before we can discuss their clinical application as iMSC-based therapy for IBD.

Reviewer 3 Report

Comments and Suggestions for Authors

The manuscript by Hammed Ayansola et al. provides valuable insights into the intricate relationship between subepithelial stromal cells and intestinal epithelial cells. However, it is noteworthy that the manuscript currently requires further adjustments and improvements before it can be considered for publication.

The following suggestions and corrections aim to enhance the overall quality and readability of the manuscript:

General Comments:

The manuscript effectively defines abbreviations like "ISC" and "TA zone" upon first use, contributing to reader understanding. However, abbreviations such as "TGFβ" and "BMPs" could benefit from being spelled out upon their initial mention to enhance clarity, especially for readers less familiar with the terminology.

Some sentences in the manuscript, particularly those discussing the roles of different cell types and molecular signaling, are quite complex. Breaking down these sentences into smaller, more digestible units would improve readability and comprehension for the reader.

While the manuscript references previous studies to support its claims, it would be beneficial to supplement these references with specific citations. This addition would enhance the credibility of the claims made and provide readers with opportunities for further exploration of the topic.

The absence of a title for Table 1 is noted. Providing a clear and descriptive title for the table would enhance its readability and help readers understand its contents at a glance.

The absence of a methods section in the provided text is notable, as it is crucial for readers to understand how the research was conducted. Including a comprehensive methods section that outlines the type of database utilized, the article's type, the interval of years studied, and key search terms would strengthen its overall impact and credibility.

The manuscript would greatly benefit from a discussion on potential limitations and avenues for future research. Proposing insightful directions for further investigation would not only enhance the manuscript's impact but also advance the field's understanding in this area.

Author Response

Dear reviewer,

Thank you for giving us the chance to address the point raised by the reviewers and submit a revised draft of the manuscript “Subepithelial stromal cells: their roles and interactions with intestinal epithelial cells during gut mucosal homeostasis and regeneration” for publication in the Biomedicines journal. Your insightful comments and those of the reviewers are used to improve the quality of the revised draft. The affected changes are highlighted in yellow. Kindly see below, in red color, for a point-by-point response to the comments and concerns of the reviewers. Also, the page number corresponds to where the changes are made in the revised manuscript.

The manuscript by Hammed Ayansola et al. provides valuable insights into the intricate relationship between subepithelial stromal cells and intestinal epithelial cells. However, it is noteworthy that the manuscript currently requires further adjustments and improvements before it can be considered for publication.

The following suggestions and corrections aim to enhance the overall quality and readability of the manuscript:

General Comments:

1: The manuscript effectively defines abbreviations like "ISC" and "TA zone" upon first use, contributing to reader understanding. However, abbreviations such as "TGFβ" and "BMPs" could benefit from being spelled out upon their initial mention to enhance clarity, especially for readers less familiar with the terminology.

Author response: Thank you for pointing out the omission. Where necessary, we have ensured that all abbreviated words are defined upon their first mention in the revised manuscript.

2. Some sentences in the manuscript, particularly those discussing the roles of different cell types and molecular signaling, are quite complex. Breaking down these sentences into smaller, more digestible units would improve readability and comprehension for the reader.

Author response: Thank you for your suggestions. We have reviewed complex sentences to improve readability and comprehension for the reader. These revisions are highlighted in yellow in the revised manuscript.

3. While the manuscript references previous studies to support its claims, it would be beneficial to supplement these references with specific citations. This addition would enhance the credibility of the claims made and provide readers with opportunities for further exploration of the topic.

Author response: We also think this is an excellent suggested. We have revised and remain most relevant references to buttress the points discussed where necessary. Revised citations are highlighted in the manuscript.

4. The absence of a title for Table 1 is noted. Providing a clear and descriptive title for the table would enhance its readability and help readers understand its contents at a glance.

Author response: Thank you for pointing out the omission. The table 1 title as been updated in the revised manuscript.

5. The absence of a methods section in the provided text is notable, as it is crucial for readers to understand how the research was conducted. Including a comprehensive methods section that outlines the type of database utilized, the article's type, the interval of years studied, and key search terms would strengthen its overall impact and credibility.

Author response: Thank you for your insightful feedback. Though the editor does not require review manuscript to include method section Review: Reviews offer a comprehensive analysis of the existing literature within a field of study, identifying current gaps or problems. They should be critical and constructive and provide recommendations for future research. No new, unpublished data should be presented. The structure can include an Abstract, Keywords, Introduction, Relevant Sections”, we have addressed your concerns in the revised manuscript. From line 57, as highlighted in yellow, we stated the search database we explored, keywords searched and the focused years to ensure the recency of the cited references – “To ensure the relevance of this review process, we explored Google scholar and NIH PubMed databases as the primary source of the referenced materials. 82% of the cited references span within 2013 and 2023. This reflect that this review was not only subjected to rigorous process, but also addressed relevant findings in recent years. Specifically, we searched keywords such as intestinal epithelial and mesenchymal cellular interactions, intestinal signaling pathways, and mesenchymal roles in inflammatory bowel diseases.”

6. The manuscript would greatly benefit from a discussion on potential limitations and avenues for future research. Proposing insightful directions for further investigation would not only enhance the manuscript's impact but also advance the field's understanding in this area.

Author response: Thank you for your insightful feedback. As highlighted in yellow from line 535, we described the process to resolve current challenges before the can ultimately proceed with the therapeutic applications of iMSC for clinical settings.

Round 2

Reviewer 3 Report

Comments and Suggestions for Authors

I went through the revised version. The authors addressed all the issues raised and made substantial changes. The manuscript has been greatly improved and is now acceptable for publication in its current form.